# The In Vitro Activity of Essential Oils against *Helicobacter Pylori* Growth and Urease Activity

**DOI:** 10.3390/molecules25030586

**Published:** 2020-01-29

**Authors:** Izabela Korona-Glowniak, Anna Glowniak-Lipa, Agnieszka Ludwiczuk, Tomasz Baj, Anna Malm

**Affiliations:** 1Department of Pharmaceutical Microbiology, Medical University in Lublin, 1 Chodzki Str., 20-093 Lublin, Poland; glowniak.lipa@gmail.com (A.G.-L.); anna.malm@umlub.pl (A.M.); 2Independent Laboratory of Natural Products Chemistry, Department of Pharmacognosy, Medical University in Lublin, 1 Chodzki Str., 20-093 Lublin, Poland; aludwiczuk@pharmacognosy.org; 3Department of Pharmacognosy, Medical University in Lublin, 1 Chodzki Str., 20-093 Lublin, Poland; tbaj@pharmacognosy.org

**Keywords:** essential oils, chemical analysis, *Helicobacter pylori*, antibacterial activity

## Abstract

The anti-*H. pylori* properties of 26 different commercial essential oils were examined in vitro by MIC (minimal inhibitory concentration) determination for the reference strain *H. pylori* ATCC 43504. We selected 9 essential oils with different anti-*Helicobacter* activities and established their phytochemical composition and urease inhibition activities. Phytochemical analysis of the selected essential oils by GC-MS method and antioxidant activity were performed. The phenol red method was used to screen the effect of essential oils on urease activity expressed as IC_50_ (the half of maximal inhibitory concentration). The most active essential oils, with MIC = 15.6 mg/L, were thyme, lemongrass, cedarwood and lemon balm oils; MIC = 31.3 mg/L—oregano oil; MIC = 62.5 mg/L—tea tree oil; MIC = 125 mg/L—pine needle, lemon and silver fir oils with bactericidal effect. Urease activity was inhibited by these oils with IC_50_ ranged from 5.3 to > 1049.9 mg/L. The most active was cedarwood oil (IC_50_ = 5.3 mg/L), inhibiting urease at sub-MIC concentrations (MIC = 15.6 mg/L). The statistical principal component analysis allowed for the division of the oils into three phytochemical groups differing in their anti-*H. pylori* activity. To summarize, the activity in vitro of the five essential oils silver fir, pine needle, tea tree, lemongrass, and cedarwood oils against *H. pylori* was found in this paper for the first time. The most active against clinical strains of *H. pylori* were cedar wood and oregano oils. Moreover, cedarwood oil inhibited the urease activity at subinhibitory concentrations. This essential oil can be regarded as a useful component of the plant preparations supporting the eradication *H. pylori* therapy.

## 1. Introduction

*Helicobacter pylori* is an etiological factor of the most frequent and persistent bacterial infection worldwide, which affects nearly half of the world’s population. *H. pylori* is recognized as the major etiological agent of including peptic ulcer disease, gastritis, gastric cancer and functional dyspepsia [1]. Moreover, extra-digestive disorders (idiopathic thrombocytopenic purpura, vitamin B12 deficiency and unexplained iron deficiency anemia) were included as indications for eradication of *H. pylori* [2]. In recent years, due to increasing antimicrobial resistance of *H. pylori*, treatment of this pathogen has remained a challenge for physicians. Currently, the first-line treatments for *H. pylori* infections are based on the combinations of a proton pump inhibitor and two antibiotics: amoxicillin and clarithromycin or metronidazole (triple therapy). As an alternative, levofloxacin can replace clarithromycin in first-line therapy, with higher cure rates [3]. Moreover, an alternative empiric strategy is mandatory when local clarithromycin resistance is higher than 20% [2]. When the triple scheme fails, a quadruple therapy containing bismuth (bismuth salts, tetracycline and metronidazole plus PPI) or non-bismuth-based quadruple therapy (i.e., levofloxacin, nitazoxanide and doxycycline plus PPI) must be recommended [2]. Treatment is justified only for symptomatic patients. Therefore, asymptomatic carriers constitute a reservoir of *H. pylori* strains in population, including antibiotic resistant strains which can easily spread. People with asymptomatic infection would benefit from a nutritional line aimed at sustaining a low level of *H. pylori* density in gastric mucosa preventing from development of severe gastritis and an increased incidence of peptic ulcer. Urease is key enzyme which enables *H. pylori* survival and colonization by initiating the hydrolysis of urea generating ammonia to neutralize stomach acid in order to create a suitable pH environment. Hence, urease is considered to be a critical target in the research and exploitation of antibacterial agents [4].

Therapeutic alternatives beyond antibiotics have been investigated in recent years, including vaccines, probiotics, photodynamic inactivation, phage therapy and phytomedicine. Several studies have been performed in the aspect of searching for plants and plant extracts/constituents as anti-*H. pylori* activity and gastroprotective action [5,6]. Moreover, components of natural origin have been extensively investigated as potential effective urease inhibitors for the treatment of *H*. *pylori* infection. Essential oils (EOs) have been shown to possess antibacterial, antifungal, antiviral, insecticidal and antioxidant properties [7]. Few articles have described the effects of specific essential oils on *H. pylori* growth and viability [8,9,10]. There is scant data on essential oils’ in vivo activity. In the study by Ohno et al. [11] the authors reported that the density of *H. pylori* in the stomach of mice treated with lemongrass was significantly reduced compared with untreated mice. In another study, by Hartmani et. al. [12], the anti-*H. pylori* activity of 2:1 mixture of *Satureja hortensis* and *Origanum vulgare* subsp. *hirtum* essential oils was investigated. In the in vivo experiments, the mixture successfully eradicated this pathogen in 70% of the mice.

The aim of this work was to identify EOs with strong antimicrobial activity against *H. pylori* growth. We examined the anti-*H. pylori* properties of 26 different commercial EOs in vitro. We selected 9 EOs with different anti-*Helicobacter* activities and established their phytochemical composition and urease inhibition activities. 

## 2. Results

### 2.1. Anti-H. Pylori Activity of Essential Oils

The results of antibacterial activity of 26 commercial EOs against reference strain of *H. pylori* ATCC 43504 are shown in Table 1. 

Based on MBC/MIC ratio < 4, it was considered that most of the tested EOs were bactericidal. Thymol, menthol and bisabolol were used as reference compounds showing strong anti-*Helicobacter* activity. For the further analysis nine of the EOs with different MIC values were chosen: silver fir (SF), pine needle (PI), tea tree (MA), lemongrass (LG), lemon (LE), lemon balm (ME), thyme (TY), oregano (OR) and cedarwood (CE) oils.

### 2.2. Analysis of Essential Oils

The volatile components detected in nine examined EOs are listed in Table 2 in order of their elution from an ZB-5MS column. Chromatographic analysis indicated considerable variability in chemical composition of these EOs. The structures of the most characteristic terpenoids found in the examined EOs are presented on Figure 1.

There are two EOs for which the monoterpene hydrocarbons are the most characteristic components. These are EOs obtained from *Pinus sylvestris* (PI) and *Citrus lemon* (LE). However, the major components for pine EO are α- (**3**) and β-pinene (**6**), belonging to bicyclic monoterpenoids, while monocyclic compound, limonene (**11**) is characteristic for lemon EO. 

In three among nine chemically analyzed EOs, monoterpene alcohols are the predominant components. Essential oils hydrodistilled from *Thymus vulgaris*, *Origanum vulgare*, and *Melaleuca alternifolia* belong to this group. Thymol (**36**), carvacrol (**37**), and terpinen-4-ol (**25**) are the dominant compounds in these EOs, respectively. Besides alcohols, the mentioned EOs are also characteristic for the presence of monoterpene hydrocarbons. The important component of thyme and oregano EOs is *p*-cymene, while other terpinenes, like α-, γ- and δ-terpinene (=terpinolene), and α-terpineol were detected in tea tree EO. 

Lemon balm EO (*Melissa* spp.) is another essential oil known for the presence of monotepene alcohols as the main compounds. Despite this similarity, chemical differences were evident between this EO and already mentioned thee oils. This difference concerns the second dominant group of monoterpenoids, which are aldehydes, not hydrocarbons. The most characteristic terpenes found in this EO are citronellal (**21**), citronellol and geraniol.

Almost 76% of all components present in the EO from lemongrass (*Cymbopogon schoenanthus*) are monoterpene aldehydes, and mainly two isomers, neral (**29**) and geranial (**33**). Geraniol and its acetate are the next two components of this EO that are worth mentioning. Another monoterpene ester, bornyl acetate (**34**) constitutes more that 53% of all volatiles present in the EO obtained from silver fir (*Abies alba*). This EO is also rich in hydrocarbons, e.g., α-, β-pinene, limonene or camphene.

It has been shown that prevailing components of the pine, lemon, silver for, thyme, oregano, lemongrass, lemon balm, and tea tree EOs are monoterpenoids. This group of terpenoids constitute about 90% of all components present in all these essential oils. Sesquiterpenoids represent a small percentage of their composition. There is a completely different situation in the case of cedarwood EO (*Juniperus virginiana*—CE). No monoterpenoids have been identified in this oil. Almost 76% of the components present in this EO are sesquiterpene hydrocarbons, while the other compounds are sesquiterpene alcohols. The most characteristic compounds found of this EO are α-cedrene (**49**) and thujopsene (**52**).

### 2.3. Antioxidant Analysis

The analyzed EOs showed poor antioxidative activity, as found on the basis of AAI (antioxidant activity index) < 0.5; the values of AAI ranged from < 0.08 to 0.114, depending on the tested oil (Table 3).

### 2.4. Urease Inhibitory Analysis

The results for the assessment of urease inhibitory activity of the EOs are listed in Table 4. It was found that the most active was cedarwood oil (IC_50_ = 5.3 mg/L), inhibiting urease at subinhibitory (sub-MIC) concentrations (MIC = 15.6 mg/L), concentration that is below the value capable of inhibiting the detectable growth and replication of a microorganism. Pine, lemon, silver fir and tea tree oils showed highly inhibitive activity for urease despite their higher MIC values. This is indicative that their subinhibitory values for bacterial growth have evident inhibitory effect against bacterial urease. 

Lemongrass, oregano, thyme oils demonstrated weaker urease inhibitory activity even though their MIC values showed strong inhibitory effects of *H. pylori* growth. Melissa oil did not inhibit the urease activity at concentrations used (IC_50_ > 1049.9 mg/L).

### 2.5. In Vitro Antimicrobial Activity of Selected Essential Oils against Clinical H. pylori Strains

The antimicrobial activity of the nine selected EOs was evaluated against 22 clinical strains including sensitive to antibiotics (10 stains) and 12 resistant strains to at least 1 antibiotic (metronidazole, rifampicin, clarithromycin, tetracycline or levofloxacin). The EOs activity was defined on the basis of MIC_50_ and MIC_90_, i.e., MIC values inhibiting 50 or 90% of the studied clinical strains, respectively. The most active were cedarwood and oregano oils with MIC_90_ = 62.5 mg/L. Less active were thyme, lemongrass and lemon balm oils (MIC_90_ = 125 mg/L), followed by tea tree, lemon, pine needle and silver fir oils (MIC_90_ = 250 mg/L). It was shown that antibiotic susceptibility of clinical *H. pylori* strains has no impact on the antibacterial activity of the tested EOs (Figure 2). 

### 2.6. Principal Component Analysis (PCA) of Phytochemical Composition and Biological Properties against H. pylori Strain of the Analyzed Essential Oils

The statistical analysis by PCA allowed to the division of the oils into three phytochemical groups differing in their anti-*H. pylori* activity (Figure 3, Table 5).

Group I included oils of similar chemical composition: silver fir, pine needle and lemon oils. The components present in these EOs belonged mainly to monoterpene hydrocarbons in pine and lemon oils. Silver fir oil also consisted of monoterpene esters (i.e., bornyl acetate), but without any influence on its bioactivity. EOs included in this group had the lowest anti-*H. pylori* and antioxidant activity but high inhibitory activity against bacterial urease. Group II consisted of EOs with quite similar chemical composition, containing: α-terpinene, *p*-cymene i γ-terpinene, but differed in the main components—carvacrol for oregano oil, terpinen-4-ol for tea tree oil and thymol for thyme oil. Bioactivity of this group was diversified. Group III included oils of different chemical composition. Main components of lemongrass and lemon balm EOs belonged to monoterpene alcohols and aldehydes in different proportion explaining the differences in their antioxidant and urease inhibitory activity. The composition of cedarwood oil was quite different from all chemically investigated EOs; the main components were sesquiterpenoids, e.g., α-, β-cedrene, thujopsene, cedrol, cuparene. This group presented the highest efficacy of *H. pylori* growth inhibition both reference and clinical strains (Table 5).

## 3. Discussion

Plant essential oils and extracts have been used for many thousands of years in food preservation, pharmaceuticals, alternative medicine and natural therapies [13]. Studies indicate that essential oils have bactericidal effects against several microorganisms [14,15,16]. However, differences in the bactericidal activity among essential oils were observed. We screened 26 different commercial essential oils for their anti-*Helicobacter* properties by determination of inhibitory and bactericidal activity. The highest activity against reference *H. pylori* strain revealed cedarwood, lemongrass, thyme, lemon balm no. 2, Ylang-Ylang and basil oils. According to bioactivity criteria established by O’Donnell et al. [17] these EOs presented strong bioactivity against *H. pylori*. The rest of the EOs exhibited good bioactivity showing MIC range of 26–125 mg/L. Out of 26 EOs 9 oils demonstrating different MIC values were chosen to display a correlation between a chemical composition and bioactivity of EO.

The activity of essential oils against *H. pylori* was reported before [11] and among the 13 essential oils tested for *H. pylori* growth inhibition, *Cymbopogon citratus* (lemongrass) and *Lippia citriodora* (lemon verbena) were demonstrated to have the highest bactericidal activities for *H. pylori* strain ATCC 49503 [18]. Bergonzelli et al. [8] evaluated the activities of 60 essential oils against *H. pylori* showing that 30 oils were able to affect the growth in vitro and 15 showed strong activity. Among compounds of these oils, carvacrol, isoeugenol, nerol, citral (=neral + geranial) and sabinene exhibited the strongest anti-*H. pylori* effects. In our study, it was shown that prevailing components of the pine needle, lemon, silver fir, thyme, oregano, lemongrass, lemon balm, and tea tree EOs are monoterpenoids. This group of terpenoids constitutes about 90% of all components present in all these essential oils. Sesquiterpenoids represent a small percentage of their composition. Terpenoids are weakly to moderately soluble in water, but they are dissolvable in the phospholipid membrane. The antimicrobial activities of terpenoids are assumed to be due to their ability to disrupt or penetrate lipid structure causing a loss of membrane integrity, dissipation of the proton motive force and impairment of intracellular pH homeostasis [19,20]. However, the different composition within the monoterpenoids group was correlated with different MIC values of EOs. In PCA group I, with the weakest anti-*H. pylori* activity among the tested EOs, monoterpene hydrocarbons were the dominating compounds, indicating barely good bioactivity against *H. pylori*. In PCA group II, despite the little difference, MIC values of EOs were higher than in group I, possibly due to the content of monoterpene alcohols. The higher content of the alcohols, the higher the MIC values that were observed. Thymol is reported to be a strong antimicrobial agent of which hydroxyl group and delocalized electrons are responsible for damaging the cytoplasmic membrane [20], interacting with membrane and intracellular proteins and changing the membrane permeability, the leakage of potassium ions and ATP [21,22] of other bacteria. Carvacrol could act by disruption and depolarization of the plasma membrane by targeting membrane proteins and intracellular drug target and was recognized as a strong inhibitive against *H. pylori*, but in the presence of thymol, the antibacterial effect of carvacrol was found to be reduced [23]. In our study, the EOs containing thymol (thyme EO) or carvacrol (oregano EO) as the main component, with the other one being found in trace amounts. Other components identified in EOs from PCA group II, including γ-terpinene, 1,8-cineole, α-pinene, β-pinene, p-cymene, have been shown, in other studies, to have no antibacterial activity against Gram negative bacteria [24,25,26,27]. Furthermore, it has been reported that some components with no antibacterial activity in the presence of some antimicrobial agents show a synergistic effect [28], for example, the presence of *p*-cymene along with carvacrol may enhance the antimicrobial activity of the oil [27]. 

The PCA group III consists of the EOs with the highest anti-*H. pylori* activity; however, for lemongrass and lemon balm EOs, the activity comes from the monoterpene alcohols and aldehydes content. In other studies, geranial and neral in lemongrass and lemon verbena EOs were marked also as components possessing heightened anti-*H. pylori* bactericidal effects compared to other components [11]. In the case of cedarwood EO (*Juniperus virginiana*), there was a completely different situation. No monoterpenoids were identified in this oil. Sesquiterpenoids are the dominant group of terpenoids present in this EO, and among them α-cedrene and thujopsene are the most characteristic compounds. There have been no reports on the biological activity of cedarwood EO and its compounds against *H. pylori* to date. However, there is published data about the anti-*H. pylori* activity of patchouli alcohol [29], which belongs to the group of tricyclic sesquiterpenoids, similarly for the major components present in cedarwood oil. It has been reported that its antibacterial activity against *Streptococcus mutans* [30] and antifungal properties are based on DNA polymerase inhibition [31]. In our study, cedarwood oil was one of the most potent EOs. 

In our study, cedarwood EO was also the most effective inhibitor of *H. pylori* urease. It might be speculated that one of sesquiterpene hydrocarbons or alcohols or their synergistic activity exhibit very strong activity against bacterial urease. Nonetheless, the activity of individual EOs components is necessary to determine. Urease inhibition may be the main defense against *H. pylori* due to prevention from adhering it to the gastric mucosa. Previous studies revealed the isolation of urease inhibitors from some plants and herbs [32,33,34]. It was found that the mechanism of urease inhibition is noncompetitive, in which both inhibitor and the substrate were attached to the enzyme non-competitively. Some flavonoids have demonstrated inhibitory effects on *H. pylori* urease [35,36,37,38]. To date, there are no studies related to analyses of essential oils towards *H. pylori* urease inhibition. Our study demonstrated that even EOs with barely good inhibitory activity against *H. pylori* growth present strong or very strong inhibitory activity against urease. This means that subinhibitory concentration of EOs could be effective in the treatment of *H. pylori* infection since urease is essential for its colonization; therefore, the inhibition of this enzyme partly explains the anti-*H. pylori* activity. 

In the radical scavenging activity assay, the tested EOs displayed weak activity. Essential oils are quite complex mixtures composed of a variety of compounds, which is why it is difficult to explain their activities. The antioxidant activity is generally related to the major active compounds in essential oils such as eugenol in clove [39], carvacrol in oregano [40], thymol in thyme [41], and citronellol or citronellal in citronella [42]. In our study, EOs included in PCA group II (thyme, oregano and tea tree EOs) presented the highest antioxidative activity. However, the other antioxidant compounds in essential oils such as terpinene, bornyl acetate, and eucalyptol have been reported to present antioxidant activity, but their amounts were probably too low to exhibit antioxidant activity [42,43]. Antioxidant activity of essential oil varies with source of essential oils and is also affected by extraction method or solvents used. Additionally, the harvesting period of the plant also determines the concentration of the major oil components such as phenolic compounds, which is directly related to the antioxidant activity of essential oils [16]. 

*H. pylori* infection preceded by gastric mucosa colonization is associated with the production of reactive oxygen and nitrogen forms. Therefore, the use of antioxidants can be regarded as a complementary therapy in *H. pylori* eradication [44]. The use of functional food and dietary supplements containing the substances with high antioxidant activity can strengthen the protective properties of the body and inhibit *H. pylori* multiplication [45]. In addition, most studies proved that *H. pylori* infection affects the level of antioxidants in gastric juice. Experimental studies, both in vivo and in vitro, have shown that substances with strong antioxidant properties such as vitamin C and astaxanthin not only scavenge free radicals, but also show antimicrobial activity against *H. pylori* [46,47].

The use of essential oils of cedarwood, oregano, lemongrass, lemon balm may contribute to an efficient control of *H. pylori* that is spread among the world population. Moreover, due to the increase of the multiresistance pattern and also to the increasing tendency of the public to consume ‘green products’, the use of these types of compounds will support treatment of *H. pylori* infection and so may help to reduce the transmission of this pathogen from asymptomatic carriers.

## 4. Materials and Methods

### 4.1. Essential Oils

The essential oils were obtained from several commercial sources available in Polish market (Table 1) and kept at 4 °C. They were tested dissolve in 100% dimethyl sulfoxide (DMSO).

### 4.2. Analysis of Essential Oils

Two μL of nine essential oils were diluted to 1 mL by *n*-hexane. One μL of the essential oil solutions was analyzed by GC-MS using a Shimadzu GC-2010 Plus gas chromatography instrument coupled to a Shimadzu QP2010 Ultra mass spectrometer (Shim-Pol, Izabelin, Poland). Compounds were separated on a fused silica capillary column ZB-5MS (30 m, 0.25 mm i.d.) with a film thickness of 0.25 μm (Phenomenex, Torrance, CA, USA). The following oven temperature program was initiated at 50 °C, held for 3 min, then increased at the rate of 8 °C/min to 250 °C, held for 2 min. The spectrometers were operated in electron-impact (EI) mode, the scan range was 40–500 amu, the ionization energy was 70 eV, and the scan rate was 0.20 s per scan. Injector, interface and ion source were kept at 250, 250 and 220 °C, respectively. Split injection (1 μL) was conducted with a split ratio of 1:20 and helium was used as carrier gas of 1.0 mL/min flow-rate. The retention indices were determined in relation to a homologous series of n-alkanes (C8–C20) under the same operating conditions. Compounds were identified using a computer-supported spectral library (NIST 2011 - Gaithersburg, MD, USA; MassFinder 2.1 and 4.0—Hamburg, Germany), mass spectra of reference compounds, as well as MS data from the literature [48,49].

### 4.3. Antioxidant Assay

Determination of 2,2-diphenyl-1-picrylhydrazyl (DPPH) activity was carried out according to the method described by Brand-Williams et al. [50] with modifications. Various dilutions of extracts (5–125 μg/mL) and butylated hydroxytoluene (BHT) (2–30 μg/mL) as a positive control were used for antioxidant assay. 100 μL of each of the tested extracts or BHT were mixed with 100 μL of ethanolic solution of DPPH (32 μg/mL) in 96 well plates. Plates were incubated in the dark at room temperature for 30 minutes after 10 seconds of shaking. Before being measured, the plates were re-shaken for 60 seconds. The absorbance was measured using a BioTek ELx808 reader (BioTek Instruments, Inc, Winooski, VT, USA) at 515 nm. The EC_50_ values (concentration at which the DPPH absorbance is reduced by 50%) were calculated by Gen5 software version 2.01 (BioTek Instruments, Inc, Winooski, VT, USA) five logistic parameters (5LP) according to the equation:Y = D + (A − D)/((1 + (x/C)^B)^E)(1)
where: *A*—minimal curve asymptote; *B*—measure of slope of curve at its inflection point, *C*—value of x at inflection time; *D*—maximal curve asymptote; *E*—quantifies the asymmetry; x concentration, *Y*—response.

The antioxidant activity index (AAI) was calculated according to the method proposed by Scherer and Godoy [51], were:AAI = final DPPH concentration (µg/mL)/EC_50_ (µg/mL)(2)

Antioxidant activity EOs was assessed according to the following scale: poor antioxidant activity when AAI < 0.5, moderate antioxidant activity when AAI between 0.5 and 1.0, strong antioxidant activity when AAI between 1.0 and 2.0, and very strong when AAI > 2.0 [16].

The assays were carried out in triplicate and all the samples and standard solutions, as well as the DPPH solutions, were prepared daily.

### 4.4. Bacteria

*H. pylori* ATCC 43504 was obtained from American Type Culture Collection (Rockville, MD, USA). The 22 clinical *H. pylori* strains used in the study were isolated from patients with gastrointestinal disorders, tested and described elsewhere [52]. Ten strains were susceptible to all tested antibiotics and out of 12 strains 4 strains were resistant to 1 antibiotic (metronidazole or rifampicin), 6 strains were resistant to 2 antibiotics (clarithromycin + metronidazole; clarithromycin + rifampicin; metronidazole + rifampicin; metronidazole + levofloxacin), 1 strain was resistant to clarithromycin + levofloxacin + metronidazole and 1 strain was resistant to clarithromycin + levofloxacin + metronidazole + rifampicin.

### 4.5. Antimicrobial Activity Testing

The strains were obtained during 72-h cultivation in the brain heart infusion agar (BHA, Becton Dickinson, Germany) + 7% horse blood in microaerophilic conditions (5% O_2_, 15% CO_2_ and 80% N_2_). Cell concentration was determined using a densitometer (BioMerieux, Marcy l’Etoile, France). Bacterial suspensions with a density of three according to the McFarland scale, i.e., 3 × 10^8^ cells (CFU)/1 mL were used for the tests. Essential oils were screened for antibacterial activities by microdilution broth method according to the European Committee on Antimicrobial Susceptibility Testing (EUCAST) (www.eucast.org) using Mueller-Hinton broth with 5% lysed horse blood. Minimal Inhibitory Concentration (MIC) of the tested essential oils were evaluated for all *H. pylori* strains with method modification by addition after incubation of resazurin to visualize the growth of *H. pylori*. Appropriate DMSO control (at a final concentration of 10%), a positive control (containing inoculum without the tested essential oils) and negative control (containing the tested essential oils without inoculum) were included on each microplate. 

Minimal bactericidal concentration (MBC) was determined by subculturing 5 μL of the microbial culture from each well that showed growth inhibition, from the last positive one and from the growth control onto the recommended agar plates. The plates were incubated at 35 °C for 72 h in microaerophilic conditions and the MBC was defined as the lowest concentration of the essential oil without growth of microorganism. Each experiment was triplicated. Representative data is presented.

### 4.6. Urease Inhibitory Effect

Urease inhibitory effect was measured via the alkalimetric method developed by Hamilton Miller and Gargan (1979) and Mobley et al. (1988) for urease preparations of *H. pylori* ATCC 43504 strain in the presence of urease inhibitors. In a 96-well plate, 10 µL of bacterial suspension (3 McFarland) was mixed with various concentrations of inhibitors (range 250–1.95 mg/L) in urea medium (0.1 g/L yeast extract, 9.1 g/L KH_2_PO_4_, 9.5 g/L Na_2_HPO_4_, 20 g/L urea and 0.01 g/L phenol red) and mixed thoroughly. After 24 h of incubation at 35 °C in microaerophilic conditions, absorbance was measured at 560 nm (BioTEK ELx808, BioTek Instruments, Inc, Winooski, VT, USA). The concentration of the inhibitor required to diminish enzyme activity by 50% (IC_50_) was calculated by plotting percent inhibition against the concentration of inhibitor (Gen5 software, BioTek Instruments, Inc, Winooski, VT, USA). One hundred percent activity was determined in the absence of inhibitor. 

### 4.7. Statistical Analysis

All analyses were performed in triplicates in order to prove their reproducibility. The results of antioxidant assays were expressed as mean ± standard deviations (SD). The representative results for antimicrobial activity testing were presented. Principal component analysis (PCA) was performed using STATISTICA 13 (StatSoft. Inc., Tulsa, OK, USA).

## 5. Conclusions

The activity in vitro of the five essential oils silver fir, pine needle, tea tree, lemongrass, and cedarwood oils against *H. pylori* was found in this paper for the first time. The most active against clinical strains of *H. pylori* were cedarwood and oregano oils. Moreover, cedarwood oil inhibited the urease activity at sub-inhibitory concentrations. This essential oil can be regarded as a useful component of the plant preparations supporting the eradication *H. pylori* therapy.

## Figures and Tables

**Figure 1 molecules-25-00586-f001:**
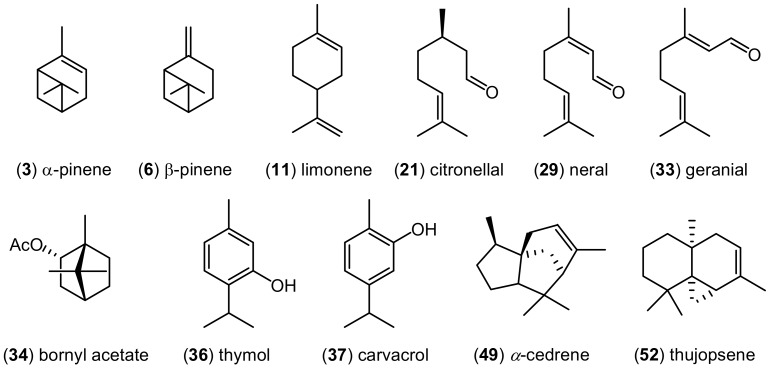
Structures of the major components present in the analyzed essential oils. Compound numbers identical to those in Table 2.

**Figure 2 molecules-25-00586-f002:**
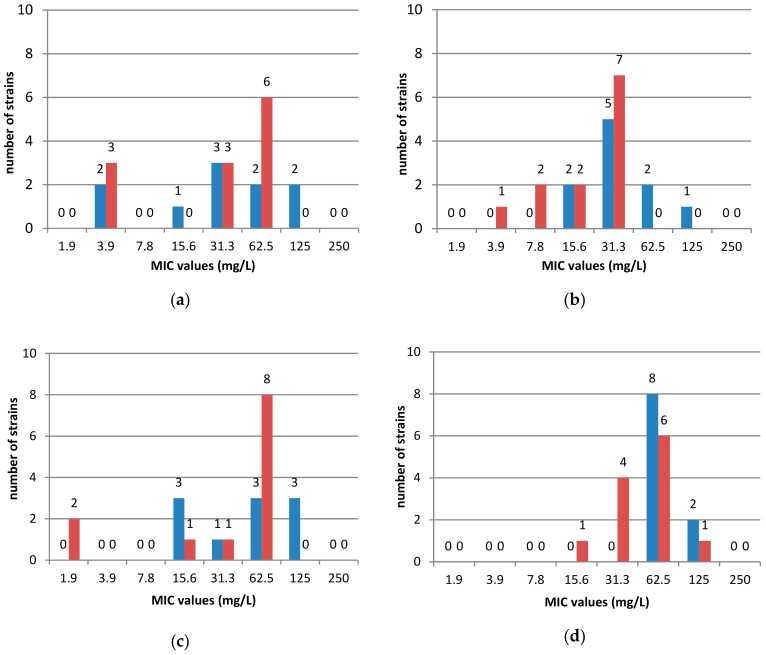
Susceptibility of 22 clinical *H. pylori* strains to essential oils in MIC values (mg/L): (**a**) cedarwood oil; (**b**) oregano oil; (**c**) thyme oil; (**d**) lemongrass oil; (**e**) lemon balm oil; (**f**) tea tree oil; (**g**) lemon oil; (**h**) pine and fir oils.

**Figure 3 molecules-25-00586-f003:**
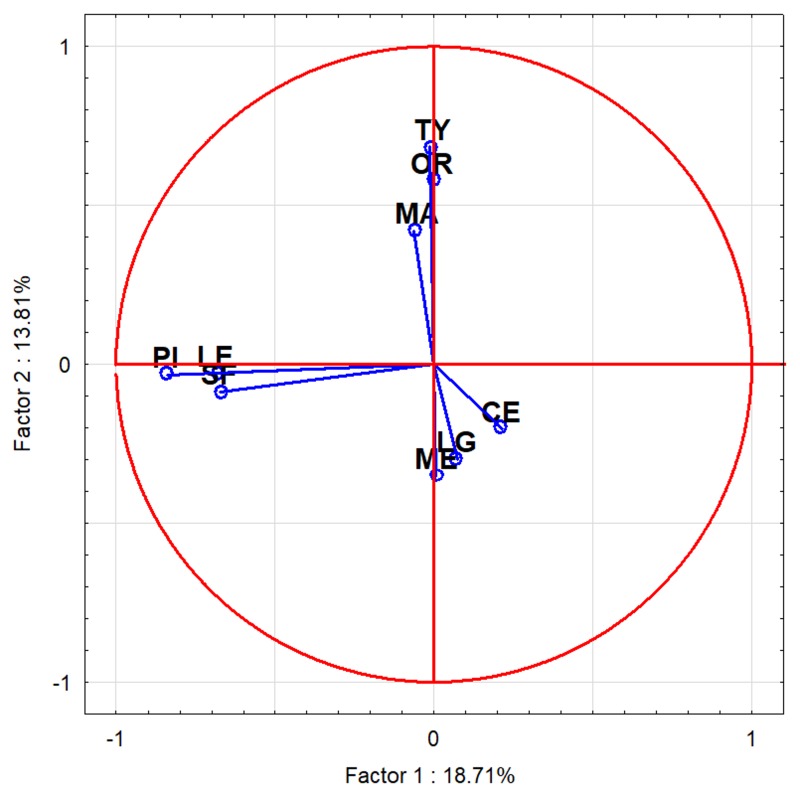
Principal component analysis of 9 analyzed essential oils based on their chemical composition. PI—pine needle EO; LE—lemon EO; SF—silver fir EO; TY—thyme EO; LG—lemongrass EO; CE—cedarwood EO; ME—lemon balm EO; MA—tea tree EO; OR—oregano EO.

**Table 1 molecules-25-00586-t001:** Activity of 26 commercial essential oils against *H. pylori* ATCC 43504.

Essential Oil Name	Plant Name, Family	Supplier	MIC (mg/L)	MBC (mg/L)	MBC/MIC Ratio
Thyme	*Thymus vulgaris* L. Lamiaceae	Avicenna-Oil	15.6	15.6	1
Lemongrass	*Cymbopogon schoenanthus* (L.) Spreng, Poaceae	Avicenna-Oil	15.6	15.6	1
Ylang-Ylang	Related to *Cananga odorata* Lam. Hook et Thomson, Annonaceae	Avicenna-Oil	15.6	62.5	1
Cedarwood	*Juniperus virginiana* L., Cupressaceae	Bamer	15.6	62.5	4
Lemon balm no. 2	*Melissa spp,* Lamiaceae	Kej	15.6	62.5	4
Basil	*Ocimum basilicum* L., Lamiaceae	Mogo	15.6	250	16
Niaouli	*Melaleuca viridiflora* Gaertn, Myrtaceae	Bamer	31.3	31.3	1
Oregano	*Origanum vulgare* L. Lamiaceae	Bamer	31.3	31.3	1
Clove	*Syzygium aromaticum* L., Myrtaceae	Bamer	31.3	31.3	1
Lemon balm no. 1	*Melissa spp,* Lamiaceae	Avicenna-Oil	31.3	62.5	2
Sandalwood	*Santalum album* L., Santalaceae	Bamer	31.3	62.5	1
Petitgrain	*Citrus aurantium* L. Rutaceae	Bamer	31.3	62.5	2
Kajeput	*Melaleuca leucadendra* L., Myrtaceae	Bamer	31.3	125	4
Tea tree	*Melaleuca alternifolia* Maiden et Betche, Myrtaceae	Avicenna-Oil	62.5	62.5	1
Rosemary	*Rosmarinus officinalis* Lindl., Lamiaceae	Avicenna-Oil	62.5	62.5	1
Geranium	*Pelargonium odorantissimum* (L.) L’Hér, Geraniaceae	Avicenna-Oil	62.5	62.5	1
Sage	*Salvia hispanica* L., Lamiaceae	Avicenna-Oil	62.5	62.5	2
Lavender	*Lavandula angustifolia* Mill., Lamiaceae	Bamer	62.5	125	2
Marjoram	*Origanum majorana* L., Lamiaceae	Dr.Beta	62.5	250	4
Peppermint	*Mentha piperita* L., Lamiaceae	Ejta	62.5	500	8
Hyssop	*Hyssopus spp.* L., Lamiaceae	Vera	62.5	500	8
Eucalyptus	*Eucalyptus globulus* Labill., Myrtaceae	Kej	62.5	500	8
Camphor	*Cinnamomum camphora* Ness et Eberm. Lauraceae	Bamer	125	125	1
Pine needle	*Pinus silvestris* L., Pinaceae	Kej	125	125	1
Lemon	*Citrus limon* (L.) Osbeck, Rutaceae	Avicenna-Oil	125	250	2
Silver fir	*Abies alba* Mill, Pinaceae	Avicenna-Oil	125	250	2
Thymol	Reference	Sigma	7.8	31.3	4
Menthol	Reference	Sigma	15.6	31.3	2
Bisabolol	Reference	Sigma	31.3	31.3	1

**Table 2 molecules-25-00586-t002:** Chemical composition of the analyzed essential oils.

No.	Compounds	RI ^a^	Chemically Analyzed Essential Oils ^b^								
			PI	LE	SF	TY	LG	CE	ME	MA	OR
1	Tricyclene	922			0.4						
2	α-Thujene	926		0.4						0.9	
3	α-Pinene	936	31.1	2.8	15.5	1.4	0.1			2.3	2.2
4	Camphene	950	1.3	0.1	4.0	0.7	0.6				0.2
5	Sabinene	973		2.3	0.2					0.3	
6	β-Pinene	978	20.4	17.0	4.6	0.3				0.7	0.1
7	Myrcene	988	1.7	1.9	0.4	1.6				0.7	1.7
8	Δ^3^-Carene	1009	15.0		5.1					0.5	
9	α-Terpinene	1017	0.1	0.1		2.3				9.8	1.1
10	*p*-Cymene	1026	1.2	1.7	0.6	22.5				3.1	14.6
11	Limonene	1030	12.1	58.1	9.1	0.8	0.2		4.4	1.0	1.1
12	1,8-Cineole	1034				0.4				4.1	0.8
13	(*Z*)- β-Ocimene	1036					0.2				
14	(*E*)- β-Ocimene	1046		0.1			0.1				
15	γ-Terpinene	1059	0.1	9.1		8.1				19.3	2.4
16	Terpinolene	1086	1.7	0.4		0.3				3.6	
17	Linalool	1101		0.2		5.5	1.0		0.8		2.6
18	(*Z*)-Limonene oxide	1135		0.1							
19	(*E*)-Limonene oxide	1139		0.1							
20	Isopinocarveol	1146	0.1								
21	Citronellal	1152					0.3		31.2		
22	(*Z*)-α-Dihydroterpineol	1154	0.2						0.4		
23	Isopulegol	1163									
24	Borneol	1178			0.6	1.0					0.5
25	Terpinen-4-ol	1187				0.6				39.6	0.2
26	α-Terpineol	1199	0.5	0.1	1.0	1.4	0.1			3.3	0.4
27	Fenchyl acetate	1219			0.2						
28	Citronellol	1227							13.9		
29	Neral	1239		1.3			32.6		0.2		
30	Carvacrol methyl ether	1242				0.2					
31	Linalyl acetate	1250				0.2					
32	Geraniol	1252					9.3		21.2		
33	Geranial	1268		2.2			42.8		0.5		
34	Bornyl acetate	1286	5.1		53.2						
35	Isobornyl acetate	1289	0.1		1.9						
36	Thymol	1301				45.4					0.5
37	Carvacrol	1310				3.8					67.7
38	Cytronellyl acetate	1346							5.4		
39	*m*-Eugenol	1355							1.1		
40	Naryl acetate	1355		0.5							
41	α-Longipinene	1356	0.1								
42	Geranyl acetate	1374		0.4			5.6		4.2		
43	Cedr-9-en	1392						0.3			
44	β-Elemene	1393					0.3		2.2		
45	204 [M]^+^, 119(100), 93(82)	1396						1.4			
46	7-*epi*-α-Cedrene	1405						0.2			
47	Longifolene	1418	0.8			0.1					
48	(*E*)-β-Caryophyllene	1426	4.2	0.4	0.7	2.6	2.5			0.3	2.9
49	α-Cedrene	1428						22.9			
50	(*E*)-β-Farnesene	1435		0.1							
51	β-Cedrene	1437						6.6			
52	Thujopsene	1447						21.8			
53	Aromadendrene	1449								1.3	
54	α-Humulene	1463			0.1	0.2	0.3		0.1		
55	*allo*-Aromadendrene	1471								0.6	
56	204 [M]^+^, 119(100), 93(53)	1476						1.0			
57	*ar*-Curcumene	1486						1.6			
58	Germacrene D	1489					0.3		1.9		
59	*cis*-β-Guaiene	1500						1.1			
60	Ledene	1501								1.3	
61	α-Muurolene	1502							0.8		
62	Bicyclogermacrene	1506								1.3	
63	β-Bisabolene	1509		0.2							
64	α-Cuprenene	1515						4.0			
65	Cuparene	1520						7.9			
66	γ-Cadinene	1520					1.3				
67	Cubebol	1524					1.0				
68	ζ-Cadinene	1526								1.4	
69	204 [M]^+^, 173(100), 119(60)	1526						3.8			
70	*trans*-Calamenene	1531						1.5			
71	γ-Cuprenene	1545						1.4			
72	Elemol	1554							3.1		
73	Carryophyllene oxide	1592		0.1	0.8	0.2	0.7				
74	Cedrol	1622			0.2			15.1			
75	12-*epi*-Cedrol	1642						0.6			
76	γ-Eudesmol	1645						0.6			
77	α-Cadinol	1656						3.1			
78	α-Eudesmol	1671						1.2			
79	β-Bisabolol	1679						0.6			
**Total**			95.8	99.7	98.6	99.6	99.3	96.7	91.4	95.4	99.0
Monoterpene hydrocarbons			84.7	94.0	39.9	38.0	1.2		4.4	42.2	23.4
Monoterpene alcohols			0.8	0.3	1.6	58.1	10.4		37.4	47.0	72.7
Monoterpene aldehydes				3.5			75.7		32.9		
Monoterpene esters			5.2	0.9	55.3	0.2	5.6		9.6		
Other oxygenated monoterpenoids				0.2		0.2					
Sesquiterpene hydrocarbons			5.1	0.7	0.8	2.9	4.7	75.5	5.0	6.2	2.9
Oxygenated sesquiterpenoids				0.1	1.0	0.2	1.7	21.2	3.1		

^a^ Retention index on ZB-5MS column; ^b^ PI—pine needle (*Pinus sylvestris*) EO; LE—lemon (*Citrus lemon*) EO; SF—silver fir (*Abies alba*) EO; TY—thyme (*Thymus vulgaris*) EO; LG—lemongrass (*Cymbopogon schoenanthus*) EO; CE—cedarwood (*Juniperus virginiana*) EO; ME—lemon balm (*Melissa officinalis*) EO; MA—tea tree (*Melaleuca alternifolia*) EO; OR—oregano (*Origanum vulgare*) EO.

**Table 3 molecules-25-00586-t003:** Antioxidative activity of selected essential oils.

Essential Oil	EC_50_ ± SD (mg/mL)	AAI
Oregano	0.70+/−0.03	0.114
Thyme	0.71+/−0.05	0.110
Lemon balm	0.78+/−0.08	0.102
Tea tree	0.93+/−0.03	0.085
Lemongrass	2.74+/−0.03	0.029
Cedarwood	5.16+/−0.10	0.015
Pine needle	>10.00	<0.08
Silver fir	>10.00	<0.08
Lemon	>10.00	<0.08

**Table 4 molecules-25-00586-t004:** Antibacterial activity (MIC) and inhibition of *H. pylori* urease (IC_50_) by analyzed essential oils.

Essential Oil	IC_50_ (mg/L)	MIC (mg/L)
Cedarwood	5.3	15.6
Pine needle	18.4	125
Lemon	35.6	125
Silver fir	37.9	125
Tea tree	39.1	62.5
Lemongrass	67.1	15.6
Oregano	208.3	31.3
Thyme	248.7	15.6
Lemon balm	>1049.9	15.6

**Table 5 molecules-25-00586-t005:** Biological activity of essential oils in relation to principal component analysis (PCA) groups.

PCA Group.	Essential Oil	MIC (mg/L)		Urease Inhibitory Activity IC_50_ (mg/L)	AAI
		*H. pylori* ATCC 43504	MIC_50/90_ for Clinical *H. pylori* Strains		
I	Silver fir	125	250/250	37.9	<0.08
	Pine needle	125	250/250	18.4	<0.08
	Lemon	125	250/250	35.6	<0.08
II	Oregano	31.3	31.3/62.5	208.3	0.114
	Thyme	15.6	62.5/125	248.7	0.110
	Tea tree	125	125/250	39.1	0.085
III	Lemongrass	15.6	62.5/125	67.1	0.029
	Lemon balm	15.6	62.5/125	>1049.9	0.102
	Cedarwood	15.6	31.3/62.5	5.3	0.015

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
