# Peer review of "The In Vitro Activity of Essential Oils against *Helicobacter Pylori* Growth and Urease Activity"

_molecules, 2020, doi:10.3390/molecules25030586_

Round 1
Reviewer 1 Report
The manuscript presents in vitro data.
Please elaborate on the significance of determination of antioxidant activity of essential oils to the determination of their anti_HP activity. What is the relationship between the two activities? Causally related or 2 separate and unrelated or only remotely related activities ?Is it absolutely necessary to determine antioxidant activity of essential oils ?
EXPLAIN IN DETAIL HOW essential oils can be used to treat patients with HP infection.
Data on anti-HP ACTIVITY IN AN ANIMAL MODEL OF HP INFECTION WOULD BE HIGHLY DESIRABLE
Is there correlation between the chemical composition of specific ingredients and anti-HP activity of an essential oil?
Author Response
Thank you for all your suggestions, we believe they helped a lot to improve our manuscript. Please find our detailed response together with summary of the manuscript changes below.
The manuscript presents in vitro data.
Please elaborate on the significance of determination of antioxidant activity of essential oils to the determination of their anti_HP activity. What is the relationship between the two activities? Causally related or 2 separate and unrelated or only remotely related activities ?Is it absolutely necessary to determine antioxidant activity of essential oils ?
Thank you very much to the reviewer for your valuable attention. The manuscript has been supplemented with literature information on the effect of substances with high antioxidant activity on diseases associated with H. pylori infection (p. 11, lines 264-272).
H. pylori infection preceded by the gastric mucosa colonization is associated with the production of reactive oxygen and nitrogen forms. Therefore, the use of antioxidants can be regarded as a complementary therapy in H. pylori eradication [Hagag et al., 2018]. The use of functional food and dietary supplements containing the substances with high antioxidant activity can strengthen the protective properties of the body and inhibit H. pylori multiplication [Chun et al., 2005]. In addition, most studies have shown that H. pylori infection affects the level of antioxidants in gastric juice. Experimental studies, both in vivo and in vitro, have shown that substances with strong antioxidant properties such as vitamin C and astaxanthin not only scavenge free radicals, but also show antimicrobial activity against H. pylori [Kang and Kim, 2017, Akynon, 2002].
Hagag, A. A.; Amin, S. M.; Emara, M. H.; Abo-Resha, S. E. Gastric mucosal oxidative stress markers in children with Helicobacter pylori infection. Infect. Disord. Drug Targets (Formerly Curr. Drug Targets Infect. Disord.), 2018,18(1), 60-67. doi: 10.2174/1871526517666170502154350.
Chun, S. S.; Vattem, D. A.; Lin, Y. T.; & Shetty, K.. Phenolic antioxidants from clonal oregano (Origanum vulgare) with antimicrobial activity against Helicobacter pylori. Process Biochem.2005, 40(2), 809-816. doi.org/10.1016/j.procbio.2004.02.018
Akyön, Y. Effect of antioxidants on the immune response of Helicobacter pylori. Clin. Microbiol. Infect. 2002, 8. 438-441. doi: 10.1046/j.1469-0691.2002.00426.x.
Kang, H. and Kim, H. Astaxanthin and β-carotene in Helicobacter pylori-induced Gastric Inflammation: A Mini-review on Action Mechanisms. J. Cancer Prev. 2017, 22(2), 57-61. doi: 10.15430/JCP.2017.22.2.57
EXPLAIN IN DETAIL HOW essential oils can be used to treat patients with HP infection.
The most effective way to use most EOs is by external application, as gargles and mouthwashes or inhalation; rarely they are used orally even if generally regarded as safe (GRAS) to ingest. In this case of oral administration they are generally diluted with milk, soy milk, or olive oil. Essential oil of J. virginiana is an approved flavouring ingredient (cedarwood oil). Within the routes of EO administration like oral intake and inhalation the nanodelivery systems encounter the mucosal lining of the nasal, lung, oral (sublingual and buccal) cavity, stomach, and gut. Nanocarriers can improve the stability of EOs against enzymatic degradation, achieve desired therapeutic levels in target tissues for the required duration with a lower number of doses, and might ensure an optimal pharmacokinetic profile to meet specific needs.
Data on anti-HP ACTIVITY IN AN ANIMAL MODEL OF HP INFECTION WOULD BE HIGHLY DESIRABLE
Thank you, we agree with your point. There is scanty data of in vivo results from essential oils. In PubMed we have found a few references:
In in vivo studies by Ohno et al. (2003) showed that the density of H. pylori in the stomach of mice treated with lemongrass was significantly reduced compared with untreated mice. [Ohno, T.; Kita, M.; Yamaoka, Y. et al. Antimicrobial activity of essential oils against Helicobacter pylori. Helicobacter 2003, 8, 207–215. doi:10.1046/j.1523-5378.2003.00146.x.]
In the study of Hartmani et. al. (2017), the anti-H. pylori activity of 2:1 mixture of Satureja hortensis and Origanum vulgare subsp. hirtum essential oils was investigated in vivo. The therapeutic efficiency was studied in a mouse model, where changes in H. pylori colonization were detected by PCR and histology of gastric samples. In the in vivo experiments, the mixture successfully eradicated the pathogen in 70% of the mice. [Harmati M, Gyukity-Sebestyen E, Dobra G, et al. Binary mixture of Satureja hortensis and Origanum vulgare subsp. hirtum essential oils: in vivo therapeutic efficiency against Helicobacter pylori infection. Helicobacter. 2017;22(2):10.1111/hel.12350. doi:10.1111/hel.12350]
There are studies describing the role of essential oils (or their compounds) in preventing peptic ulcer disease in vivo, however the model of PUD was induced by use of ethanol in animals:
Bonamin F, Moraes TM, Dos Santos RC, et al. The effect of a minor constituent of essential oil from Citrus aurantium: the role of β-myrcene in preventing peptic ulcer disease. Chem Biol Interact. 2014;212:11–19. doi:10.1016/j.cbi.2014.01.009;
Rozza AL, de Mello Moraes T, Kushima H, Nunes DS, Hiruma-Lima CA, Pellizzon CH. Involvement of glutathione, sulfhydryl compounds, nitric oxide, vasoactive intestinal peptide, and heat-shock protein-70 in the gastroprotective mechanism of Croton cajucara Benth. (Euphorbiaceae) essential oil. J Med Food. 2011;14(9):1011–1017. doi:10.1089/jmf.2010.0173;
Ribeiro AR, Diniz PB, Pinheiro MS, Albuquerque-Júnior RL, Thomazzi SM. Gastroprotective effects of thymol on acute and chronic ulcers in rats: The role of prostaglandins, ATP-sensitive K(+) channels, and gastric mucus secretion. Chem Biol Interact. 2016;244:121–128. doi:10.1016/j.cbi.2015.12.004.
We have added information regarding animal study to the Introduction together with the references (p. 2, lines 65-70).
Is there correlation between the chemical composition of specific ingredients and anti-HP activity of an essential oil?
Among the EOs tested in our study, the best anti-HP activity showed cedar wood EO. As it was mentioned, the chemical composition of this EO is very different from the other analyzed oils. The major components are tricyclic sesquiterpenoids, α-cedrene, thujopsene and cedrol. These compounds make up almost 60% of all volatiles present in the EO, and they could be responsible for the activity. There is no data so far concerning the anti-HP activity of cedar wood EO and its compounds, however one can find the data about anti-HP activity of patchouli alcohol (Xu et al., 2017) which belong to the group of tricyclic sesquiterpenoids, as like as the major components present in cedar wood oil. This information was added to the discussion (p. 10, lines 234-236).
Xu Y.F., Lian D.W., Chen Y.Q., Cai Y.F., Zheng Y.F., Fan P.L., Ren W.K., Fu L.J., Li Y.C., Xie J.H., Cao H.Y., Tan B., Su Z.R., Huang P., In Vitro and In Vivo Antibacterial Activities of Patchouli Alcohol, a Naturally Occurring Tricyclic Sesquiterpene, against Helicobacter pylori Infection. Antimicrob. Agents Chemother., 2017, 61: e00122-17
Reviewer 2 Report
The inhibition of essential oils on H. pylori and urease activity were evaluated, and the components were analysed using GC-MS. This manuscript can be accepted for publication. Some comments: 1. What is EOs in abstract? 2. "sub-inhibitory concentrations" is what? please specify. 3. In table 1, SD should be added.
Author Response
Thank you for all your suggestions, we believe they helped a lot to improve our manuscript. Please find our detailed response together with summary of the manuscript changes below.
The inhibition of essential oils on H. pylori and urease activity were evaluated, and the components were analysed using GC-MS. This manuscript can be accepted for publication.
Some comments:
What is EOs in abstract?
It was explained.
"sub-inhibitory concentrations" is what? please specify.
Concentration that is below one capable of inhibiting the detectable growth and replication of a microorganism. It was added to the text (p. 7, lines 137-138).
In table 1, SD should be added.
According to EUCAST and CLSI standards the MIC/MBC values are presented without standard deviation. Even though we did the antimicrobial activity testing three times for reference H. pylori strain for each EO (Table 1), representative data is presented. If it comes to clinical H. pylori strains the MIC ranges and MIC values inhibiting 50 or 90% of the studied clinical strains are presented. The manuscript was modified accordingly, with all the missing information added (p.13, lines 343-344, 357-359).
Round 2
Reviewer 1 Report
The revised manuscript is acceptable for publication.